# Demodulation of Three-Phase AC Power Transients in the Presence of Harmonic Distortion

**Benjamin T. Gwynn *** and **Raymond de Callafon**

Department of Mechanical and Aerospace Engineering, University of California San Diego,
La Jolla, CA 92092, USA; callafon@ucsd.edu
*   Correspondence: btgwynn@hotmail.com

**Abstract:** Load switches in power systems may cause oscillations in active and reactive power flow. Such oscillations can be damped by synthetic inertia provided by smart inverters providing power from DC sources such as photovoltaic or battery storage. However, AC current provided by inverters is inherently non-sinusoidal, making measurements of active and reactive power subject to harmonic distortion. As a result, transient effects due to load switching can be obscured by harmonic distortion. An RLC circuit serves as a reference load. The oscillation caused by switching in the load presents as a dual-sideband suppressed-carrier signal. The carrier frequency is available via voltage data but the phase is not. Given a group of candidate signals formed from phase voltages, an algorithm based on Costas Loop that can quickly quantify the phase difference between each candidate and carrier (thus identifying the best signal for demodulation) is presented. Algorithm functionality is demonstrated in the presence of inverter-induced distortion.

**Keywords:** demodulation; harmonic distortion; costas loops; pulse width modulated inverters; power system control

---

## 1. Introduction

With the advent of integration of renewable energy resources, more fluctuations and transient effects are observed on the electric grid [1,2]. Operators have traditionally had no operational option to counteract them while in progress because of their short-lived nature and slow response of conventional AC generators [3]. Direct action is not always necessary though, because the large spinning mass of conventional generators possesses considerable rotational inertia, serving to help stabilize the grid [4]. However, the ever expanding penetration of renewable distributed generation counteracts this effect, as it is predominantly in the form of photovoltaic (PV) cells with little inertia [5]. As a greater portion of power provided to the grid comes from PV there is less rotational inertia per unit of energy distributed. This is of utmost concern as transients stress power systems and can result in cascading effects [6] and widespread damage [7].

Despite the low inertial effect, modern inverters could be programmed or controlled to simulate synthetic inertia [8]. Examples include droop control emulating the generator response [9,10]. Unfortunately, current supplied by typical PV inverters is inherently non-sinusoidal and may have multiple higher harmonics, distorting power quality and the ability to measure power fluctuations accurately. Such harmonics are caused by the use of square waves, modified sine waves or pulse width modulation (PWM) to create an AC current [11].

An example of the distortion in AC current of a grid-tied inverter is illustrated in Figure 1, where measurements of voltage and current waveforms are taken at 256 samples/cycle over a resistive and inductive load. The measurements are taken at the Synchrophasor Grid Monitoring and Automation (SyGMA) Lab at the University of California, San Diego (UCSD) and a comparison is

made between the constituent current and voltage waveforms when the grid supplies power to the load (top) and when the inverter supplies power to the load and grid (bottom). It is clear that the inverter creates several harmonics on the current and it will be harder to qualify the active and reactive power flow produced by the inverter. The relationship between inverter operation and harmonic effects is discussed in Reference [12].

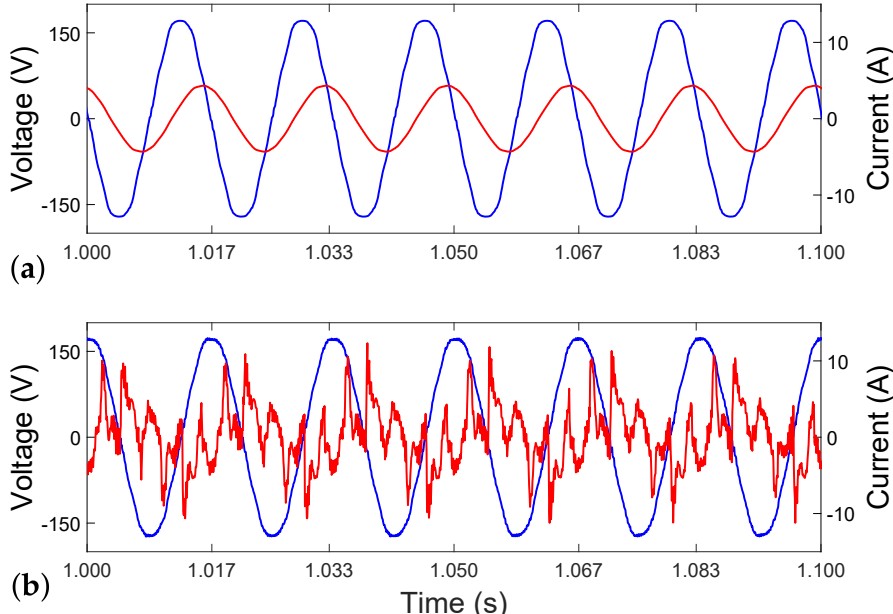

**Figure 1.** Voltage (blue) and current (red) wave forms when the inverter (**a**) is off and (**b**) supplies 1.125 kW of active power to the load and grid, illustrating significant harmonic distortion on inverter current.

Despite the harmonic distortion that may be introduced by a smart inverter, the aim is to use the inverter to provide synthetic inertia to controlled active and reactive power in order to mitigate transient power oscillations. Such mitigation is typically done via feedback of measured power oscillations to the power demand signals sent to the inverter [13]. Inverter power output modulation via feedback control can emulate droop control [14,15], and novel schemes have been a topic of increasing interest in terms of local [16,17], centralized [18], and distributed control [19,20]. Clean power measurements are necessary to be used for feedback.

Active damping for the purpose of improving power quality from PWM inverters with inductor-capacitor-inductor (LCL) filters has been discussed in [21–23], among many others. Alternatively, multistep inverters [24] or use of control with inherent damping characteristics [25] can also greatly reduce harmonics. Utility grid standards such as those in Reference [26] motivate much of this research, but the widespread use of less-sophisticated inverters is often overlooked [27]. A control approach that works in the presence of significant harmonic distortion would allow the use of legacy inverters. Development of a method to extract high quality power flow signals and provide real-time damping control to these inverter is more practical.

One obvious way to eliminate the effect of higher harmonics is simple filtering of voltage and current signals. However, such filtering may influence and eliminate the measurement of power fluctuations or adversely impact the feedback control needed to mitigate power oscillations by the inverter.

Parameter uncertainty inherent in power systems dictates use of a controller utilizing the internal model principle [28], which can be implemented on a microprocessor as shown in Reference [13]. However, demodulation of power is necessary for real time control [29] due to the response time of the microprocessor. As such, processing of three phase voltage and current signals with harmonics must be a trade-off between filtering and power oscillation demodulation.

Given the trade-off between filtering of harmonics and the need to demodulate transient power oscillations, this paper describes an algorithm that extracts active and reactive power fluctuations in the presence of distortion. The acquired signal may then be used for control of a smart inverter to mitigate transient oscillations. The algorithm is a modification to Costas Loop to demodulate active and reactive power without need of a priori knowledge about the disturbance. Demodulation in the context of power calculations has been explored extensively for use in determining voltage frequency in Phasor Measurement Units [30–32], but limitations in the method for demodulating transients presented in Reference [29] motivates the research presented in this paper.

The demodulation approach from Reference [29] makes an unconventional use of the Clarke transformation, mapping active power into the Clarke domain. Due to squaring and a subsequent square root operation, that algorithm provides accurate demodulation only when a transient initiates while both $\alpha$ and $\beta$ Clarke components of active power are positive. An experimental setup can be contrived to assure this condition is met, but variations in grid parameters common in day-to-day operation make the algorithm unfit for general usage.

The contribution of this paper is the formulation of a decoding algorithm characterizing real and reactive power flow even in the presence of large harmonic distortions as illustrated in Figure 1. The novel decoding algorithm presented in this paper allows characterization of the transient effect in power flow when loads are switched in without explicit knowledge of the dynamic parameters of the loads. Furthermore, unlike in Reference [29], reactive power is demodulated concurrently. Clearly, with better information on active/reactive power, control applications that use such active and reactive power measurements will also benefit. Inverter actuated reactive power compensation accomplished using the decoding algorithm is presented in Reference [33], while this paper is presented to convey how the decoding is accomplished.

The remainder of the paper is organized as follows—Power flow through a reference circuit is discussed forming ground truth for simulation and testing; existing demodulation methods and their limitations are presented; the signal processing involved in the novel decoding algorithm is described; the effect of harmonic distortion on the algorithm is analyzed; finally, to illustrate the effectiveness of the algorithm proposed in this paper, the modified Costas Loop is implemented on a field-programmable gate array (FPGA) for real time processing and tested on power oscillations induced by switching in a three-phase reference circuit in the test apparatus depicted in Figure 2.

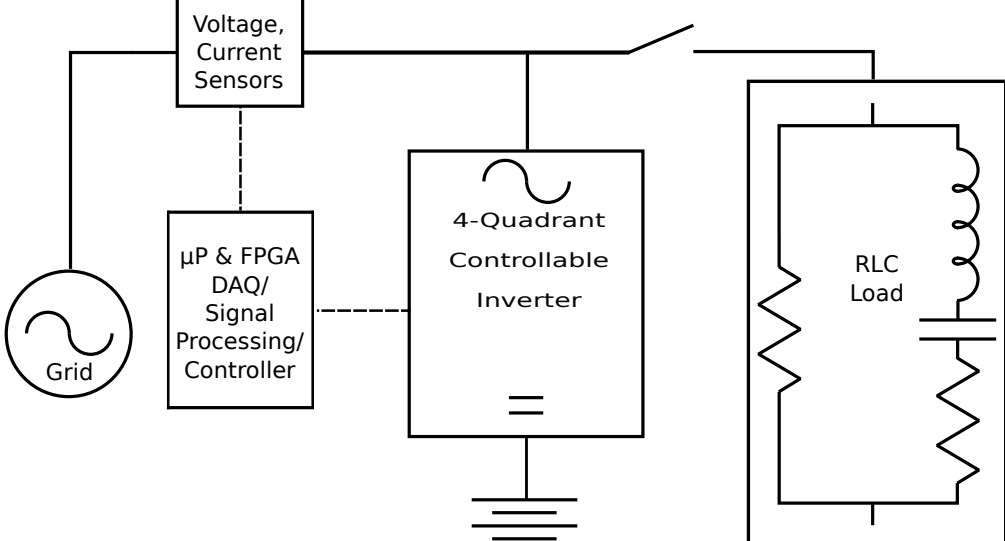

**Figure 2.** Test system configuration of inverter, grid, load, sensors, and microcontroller implemented at SyGMA Lab at UCSD.

The three-phase reference circuit is an RLC load that produces known oscillations in both active and reactive power and will serve as a reference to the actual demodulated active and reactive power.

## 2. AC Power Flow and Demodulation

### 2.1. Reference Circuit

Transient data from a live electric grid in response to a major disturbance such as a branch trip is sparse because such incidents are unplanned and most utilities use supervisory control and data acquisition systems sampling only every 2–4 s [34]. Such transient data is irreproducible due to utility obligations and customer expectations of continuity of service. However, any distribution electric grid of reasonable size will have a combination of inductive, capacitive and resistive elements, making a series RLC circuit a reasonable reference point for analysis of power flow during a transient. An RLC circuit allows design parameter selection such that transient response remains within limits of protective devices, minimizing impact on other grid customers while enabling laboratory recreation of fluctuations qualitatively similar to those witnessed on the local microgrid as shown in References [13,29].

Consider Figure 3, depicting a circuit with an ideal RLC series branch parallel with a purely resistive branch. This circuit serves as a reference in computer simulations and an actual physical implementation, at the SyGMA lab. Simulation of the reference circuit is used as a baseline for the power oscillations to be demodulated. The actual three-phase RLC circuit at the SyGMA lab is equipped with a smart inverter to provide realistic AC signals with possible distortion. Circuit parameters are as follows: $v(t)$ = 120 V RMS AC at 60 Hz, $R_1$ = 100 $\Omega$, $R_2$ = 2 $\Omega$, $L$ = 0.1 H, $C$ = 0.01 F. Note that $L$ and $C$ are higher than typical electric grid values, but were selected to allow reproduction of fluctuations with greater amplitude than otherwise possible on a low voltage circuit.

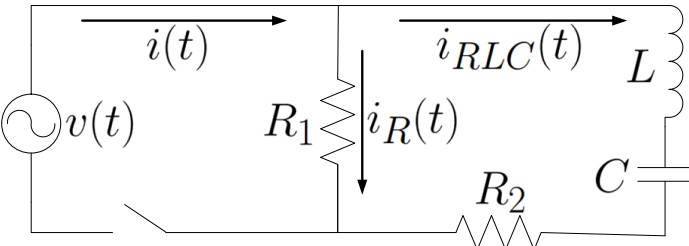

**Figure 3.** Reference circuit for creating a power oscillation.

For $t \geq t_s$ (the time of switch closure) with $v(t) = V \cos(\omega_c t)$, current response is governed by an ODE with general solution

$$i(t) = I_p \cos(\omega_c t - \alpha) \tag{1}$$

$$+ I_h \cos(\phi_s) e^{-\eta(t-t_s)} \cos(\omega_0(t - t_s) - \beta_\phi(t_s)), \tag{2}$$

where (1) is the particular solution and (2) is the homogeneous solution.

In 3-phase, consider $v_a(t) = v(t) = V \cos(\omega_c t))$, $v_b(t) = V \cos(\omega_c t + \frac{2\pi}{3})$ and $v_c(t) = V \cos(\omega_c t - \frac{2\pi}{3})$. Expressions for phase currents are omitted for brevity, noting each will be composed of the homogeneous and particular ODE solutions given respective initial conditions. Instantaneous active power in three-phase circuits can be calculated [35] as follows:

$$P(t) = v_a(t)i_a(t) + v_b(t)i_b(t) + v_c(t)i_c(t)$$
$$= P_p(t) + P_h(t). \tag{3}$$

where

$$P_p(t) = \frac{3}{2} V I_p \cos(\alpha),\qquad(4)$$

and

$$P_h(t) = \frac{3}{2} V I_h e^{-\eta(t-t_s)} \cos(\omega_0(t-t_s) - \beta) \cos(\omega_c t - \phi_s),\qquad(5)$$

making $P(t)$ independent of the time of switch closure.

As seen in Equation (5), the transient power flow $P_h(t)$ is made up of three components: exponential decay multiplied by a sinusoid at grid angular frequency $\omega_c$ and another sinusoid at the circuit natural frequency $\omega_0$, much lower than the grid frequency in practice [36] and by design in the reference circuit.

Consensus on a comprehensive theory of instantaneous reactive power in three-phase systems has not been reached, as illustrated by disagreements between References [37–39] among others. Disagreements notwithstanding, the equation for instantaneous reactive power

$$Q(t) = \frac{1}{\sqrt{3}} \Big[ v_{bc}(t) i_a(t) + v_{ca}(t) i_b(t) + v_{ab}(t) i_c(t) \Big],\qquad(6)$$

in a balanced three-phase system presented in [35] holds.

It can be shown that in a balanced circuit the product of phase currents and line voltages is equivalent to that of phase currents and phase voltages with a $-\pi/2$ radian phase shift (denoted by superscript '$-90$') and scaling factor of $\sqrt{3}$, yielding:

$$Q(t) = v_a^{-90}(t) i_a(t) + v_b^{-90}(t) i_b(t) + v_c^{-90}(t) i_c(t),\qquad(7)$$

consistent with theorems in Reference [37] showing reactive current to be orthogonal to voltage. This orthogonality will be leveraged later.

### 2.2. Power Demodulation

The power flow fluctuation $P_h(t)$ in (5) can be revealed by sampling at $t = \frac{n}{f_c} + \frac{\phi_s}{2\pi}$, insuring $\cos(\omega_c t - \phi_s) = 1$. Unfortunately, without knowledge of $\phi_s$ and the timing of switch closure, it is not realistic.

Figure 4 shows simulations of reference circuit active power after switch closure with high-speed sampling at $256 f_c$, and sampling at grid-frequency $f_c$ with sample timing varied by a quarter cycle. The figure illustrates the unacceptable changes in the signal caused by timing variations, comparing sampling timed to perfectly capture the signal envelope to sampling that almost entirely fails to capture the transient dynamics.

In (5), the amplitude term $\frac{3}{2} V I_h e^{-\eta(t-t_s)} \cos(\omega_0(t-t_s) - \alpha)$ acts on $\cos(\omega_c t - \phi_s)$. By trigonometric identity

$$\cos(\theta_1)\cos(\theta_2) = \frac{1}{2}\Big[ \cos(\theta_1 + \theta_2) + \cos(\theta_1 - \theta_2) \Big]\qquad(8)$$

the frequency content is concentrated at $\omega_c + \omega_0$ and $\omega_c - \omega_0$, making simple low-pass filtering ineffective for isolating the fluctuation at reference circuit natural frequency $\omega_0$.

A promising alternative is Amplitude Modulation (AM), as AM tools used for radio communications are faced with the same challenge in reconstructing signal envelope. In AM terms, the low-frequency component of active power transient response modulates a grid-frequency carrier in a dual sideband, suppressed carrier (DSB-SC) signal [40].

However, radio communications carriers are much higher frequency and transient effects in demodulation exhibited over several carrier cycles are therefore very short lived and imperceivable to the listener [41]. Control efforts for power systems with a carrier frequency of 50 or 60 Hz cannot afford several cycles as power transient effects must be controlled in real-time. This paper explores the possibility of adapting AM tools to demodulate active and reactive power flow fluctuations.

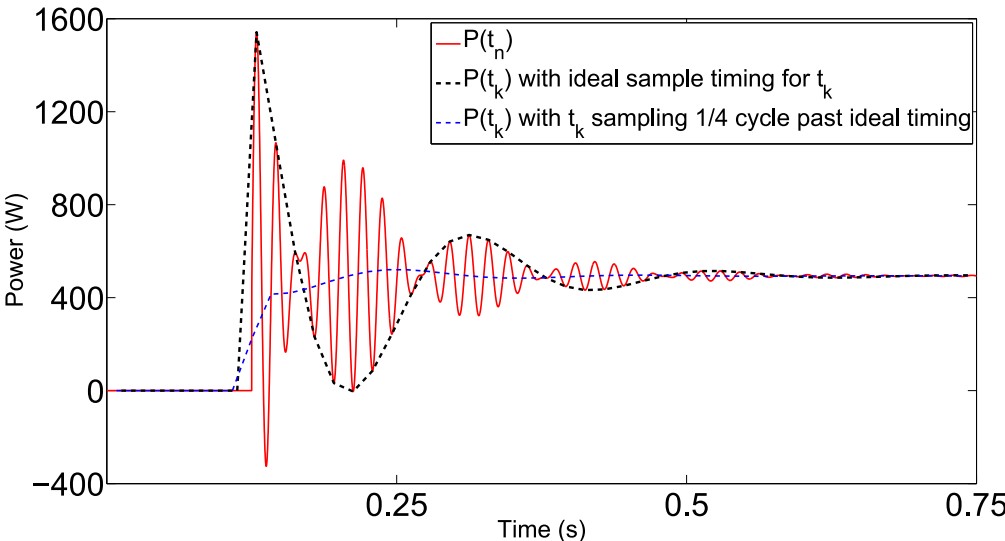

**Figure 4.** Effect of sample timing, illustrating the correct power envelope only when sampled at the correct timing relative to start of power disturbance.

The effect of harmonics in the AC current provided by a PWM inverter illustrated earlier in Figure 1 must also be taken into account. Without careful design, simple low-pass filtering to remove harmonic distortion may also remove the relevant power oscillation to be demodulated. It also introduces a time delay in the power measurement signal as indicated in Figure 5, comparing measured inverter-supplied three-phase active power $P(t_n)$ upon switch closure and the same signal low-pass filtered, denoted $\hat{P}(t_n)$. Such filtering is straightforward to implement, but requires careful consideration of trade-offs between delay and filter performance.

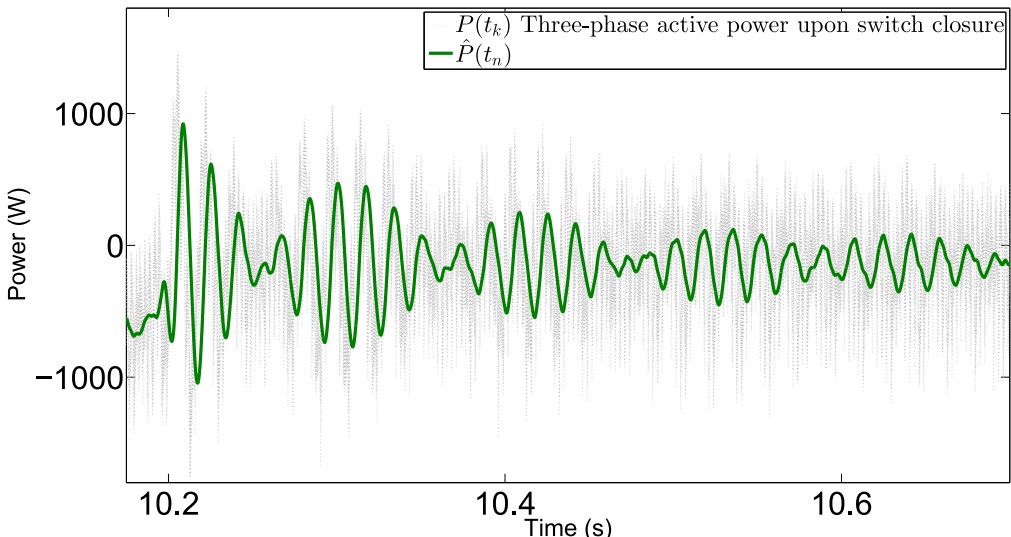

**Figure 5.** Actual real-time measured three-phase active power compared to the filtered three-phase active power.

## 3. Demodulation via Costas Loop

Drawing similarities between AM communications signals and power system fluctuations, we first present Costas Loop [41] used for demodulation of DSB-SC signals.

A DSB-SC signal exhibits $\pi$ radian phase-shifts at zero crossings of the information signal [42], preventing demodulation with a phase-locked loop (PLL) [43]. Costas loop overcomes this limitation by combining two PLLs, denoted in-phase and quadrature PLLs, as shown in Figure 6. In Figure 6 $u(t)$ is a DSB-SC input, $Q_c(t)$ is the quadrature branch and $I_c(t)$ is the in-phase branch as well as

output. The $-90°$ block indicates a $-\pi/2$ radian phase shift, LPF is a low-pass filter, VCO is a Voltage Controlled Oscillator and $\otimes$ denotes multiplication.

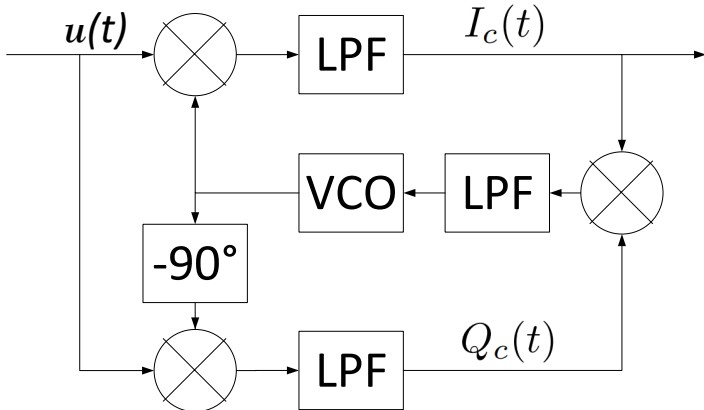

**Figure 6.** Classical Costas Loop for demodulating a dual sideband, suppressed carrier (DSB-SC) signal $u(t)$.

Costas Loop 'locks' when the phase difference between the carrier and the in-phase PLL is either 0 or $\pi$ radians [44] making phase ambiguity an issue if signal inversion is significant. There are three main reasons why Costas Loop cannot be applied directly to demodulation of power signals:

- Costas loop exhibits transient effects itself, and these transient effects interfere with the desired demodulation of power flow transient effects.
- Costas loop demodulates the amplitude with an ambiguous sign: the phase of the demodulated signal may be misread by $\pi$ radians, causing a sign mismatch in the demodulated signal.
- The magnitude of the feedback signal is dependent on that of the input; the VCO sensitivity may be optimal for one transient while completely ineffectual for another.

These effects are detrimental if the demodulated signal is to be used for real time control to suppress or dampen power oscillations. The standard solution to phase ambiguity is sending a test signal and checking for inversion [45], not viable for power fluctuations. The following section describes a modification to Costas Loop that overcomes these limitations, hereafter referred to as the 'Gwynn Open-Loop Demodulation' or GOLD method.

## 4. The GOLD Method

### 4.1. Rationale

Voltage acts as a forcing function of active and reactive power transients. The carrier frequency is the voltage frequency, and as such a VCO, as used in Costas Loop of Figure 6 is not needed. The power calculation used in Reference [29] recognized that each voltage signal could be normalized, making local generation of a demodulation signal unnecessary.

Figure 7 is a block diagram of the GOLD method algorithm. Voltage normalization occurs at position ① in the figure, where A is unitizing gain. In Figure 7 BPF is a band-pass filter, MA is a moving average filter, SIGN is a sign function. The figure shows wave forms of the respective signals as they undergo signal processing operations, and an asterisk (*) denotes that the waveform illustration time axis is subject to 6x decrease in scale compared to the other wave forms.

We introduce sampling $t_n$ and normalized voltage signals $(\tilde{v}_m(t_n) = v_m(t_n)/V)$ for demodulation. Here, subscript $m$ indicates signals formed from voltage data such as $v_a(t_n)$, $v_b(t_n)$, and $v_c(t_n)$ or delayed versions of the same. These signals will be referred to as 'candidates' with the aim of determining which candidate best demodulates $P_h(t_n)$ in terms of the gain of the demodulated power $P_G(t_n)$. This allows deviation from the operating concept of Costas Loop (as a closed-loop system in

which feedback adjusts phase and frequency to match the carrier) to a battery of open-loop calculations on demodulation signal candidates.

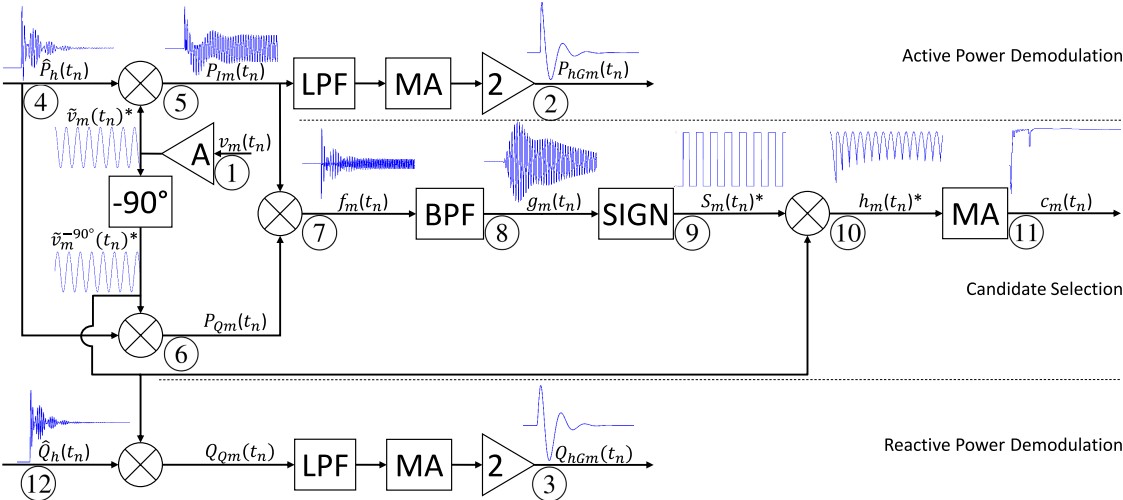

**Figure 7.** The GOLD method algorithm.

The GOLD method algorithm accomplishes three functions: Given generic candidate with in-phase signal $\tilde{v}_m(t_n) = \cos(\omega_c t - \phi_m)$ and quadrature signal $\tilde{v}_m^{-90}(t_n) = \sin(\omega_c t_n - \phi_m)$, the top ②and bottom ③branches of Figure 7 demodulate and filter active and reactive power, respectively. The middle section of the figure quantifies the phase difference between $\tilde{v}_m(t_n)$ and the fluctuation carrier, the $\cos(\omega_c t_n - \phi_s)$ term from (5).

*4.2. Algorithm*

Demodulation operations remove DC components, so these are reconstructed by

$$\hat{P}_p(t_n) = F_{MA}(q)F_{LP0}(q)P(t_n), \tag{9}$$

where $F_{MA}(q)$ and $F_{LP0}(q)$ represent transfer functions of a moving average filter and low pass filter, respectively.

$P_h(t_n)$ is the part of $P(t_n)$ requiring demodulation. It is approximated by

$$\begin{aligned} \hat{P}_h(t_n) &= \hat{P}(t_n) - \hat{P}_p(t_{n-1}), \\ &= D(q)P_h(t_n) + B(t_n), \end{aligned} \tag{10}$$

where $D(q)$ represents delay due to filtering $P(t_n)$ to get $\hat{P}(t_n)$ and $B(t_n)$ represents the error introduced by imperfections in filtering to get $\hat{P}(t_n)$ and $\hat{P}_p(t_n)$. The use of $\hat{P}_p(t_{n-1})$ facilitates parallel operations on the FPGA. $\hat{P}_h(t_n)$ is the input signal to the algorithm at ④in Figure 7.

Applying (8) to the top/in-phase branch at ⑤, the demodulated active power in the branch is given by $P_{Im}(t_n) = \hat{P}_h(t_n)\tilde{v}_m(t_n)$. Combining (5) and (10) and the definition of $\tilde{v}_m(t_n)$, $P_{Im}(t_n)$ can be expanded as

$$P_{Im}(t_n) = B(t_n)\cos(\omega_c t_n - \phi_m) \tag{11}$$

$$+D(q)\Big[\frac{3}{2}VI_h e^{-\eta(t_n - t_s)}\cos(\omega_0(t_n - t_s) - \beta)\frac{1}{2}[\cos(2\omega_c t_n - \phi_s - \phi_m) + \cos(\phi_m - \phi_s)]\Big]. \tag{12}$$

At ⑥, the quadrature branch $P_{Qm}(t_n) = \hat{P}_h(t_n)\tilde{v}_m^{-90}(t_n)$ could be expanded similarly.

The fluctuation (12) is approximated at ②by

$$P_{hGm}(t_n) = 2 \cdot F_{MA}(q)F_{LP}(q)P_{Im}(t_n), \tag{13}$$

where a moving average filter and low pass filter remove the $\omega_c$ and $2\omega_c$ frequency sinusoids of (11) and (12), respectively. Gain of 2 counteracts coefficient $\frac{1}{2}$ in (8).

Clearly (12) has greatest magnitude when $\phi_m = \phi_s$. For best demodulation we must identify the candidate with phase closest to meeting this condition. This is the purpose of the candidate selection section of the GOLD method algorithm, made up of points ⑦–⑪ of Figure 7.

Relationship $\sin(\theta) = \cos(\theta - \pi/2)$ coupled with (8) allows $\tilde{v}_m(t_n)\tilde{v}_m^{-90}(t_n){=}\frac{1}{2}\sin(2\omega_c t_n - 2\phi_m)$. Then at ⑦

$$f_m(t_n) = P_{Im}(t_n)P_{Qm}(t_n)$$
$$= \frac{1}{2}\hat{P}_h^2(t_n)\sin(2\omega_c t_n - 2\phi_m). \tag{14}$$

Expanding $\hat{P}_h^2(t_n)$ by inserting (5) into (10), successive multiplication operations lead to an inflation of frequencies. Higher frequency terms will be omitted for brevity, but the cross terms from $\hat{P}_h^2(t_n)$ lead to the retention of one important component:

$$f_m(t_n) = B(t_n)\frac{3}{2}VI_hD(q)\left[e^{-\eta(t_n-t_s)}\cos(\omega_0(t_n-t_s)-\beta)\cos(\omega_c t_n - \phi_s)\sin(2\omega_c t_n - 2\phi_m)\right] \tag{15}$$
$$+ \text{[omitted higher-frequency terms]}.$$

Using (8) again, the product of sinusoids at frequencies $\omega_c$ and $2\omega_c$ in (15) is equal to $\frac{1}{2}[\sin(\omega_c t_n - 2\phi_m + \phi_s) + \sin(3\omega_c t_n - 2\phi_m - \phi_s)]$. The $\sin(\omega_c t_n - 2\phi_m + \phi_s)$ term is the key to calculating which candidate is closest in phase with the fluctuation carrier, the $\cos(\omega_c t - \phi_s)$ term from (5). This term is isolated through band-pass filtering of $f_m(t_n)$ at ⑧:

$$g_m(t_n) = F_{BP}(q)f_m(t_n)$$
$$= G(t_n)\sin(\omega_c t_n - 2\phi_m + \phi_s) \tag{16}$$

where $G(t_n)$ is introduced to denote the unwanted dynamic effects of the band-pass filter.

The effect of $G(t_n)$ is minimized by subjecting $g_m(t_n)$ to a sign function, forming square wave $S_m(t_n) = \text{sgn}(\sin(\omega_c t_n - 2\phi_m + \phi_s))$ at ⑨.

$S_m(t_m)$ is multiplied by $\tilde{v}_m^{-90}(t_n)$ at ⑩. The resulting signal can be rearranged as

$$h_m(t_n) = |\tilde{v}_m^{-90}(t_n)|\text{sgn}\left[\tilde{v}_m^{-90}(t_n)\sin(\omega_c t_n - 2\phi_m + \phi_s)\right]. \tag{17}$$

This shows that $h_m(t_n)$ is a bounded, periodic function. Once more using (8) on the argument of the sign function in (17) yields $\frac{1}{2}(\cos(\phi_m - \phi_s) - \cos(\omega_c t_n - 3\phi_m + \phi_s))$, which is positive for the greatest part of its cycle when $\phi_m - \phi_s$ is minimized.

At ⑪ a moving average of $h_m(t_n)$ provides $c_m(t_n) = F_{MA}(q)h_m(t_n)$, a metric inversely proportional to the difference $\phi_m - \phi_s$.

Defining $\tilde{v}_G$ as the candidate with greatest $|c_m(t_n)|$ value, the corresponding $P_{hGm}(t_n)$ is selected as the algorithm output. If $c_m(t_n)$ is negative, the output is inverted, making use of symmetry of demodulated signals.

The algorithm output is added to $\hat{P}_p(t_n)$ forming $P_G(t_n)$, the best approximation of $P(t_n)$ which will be the controller input.

The calculation of reactive power in the bottom branch starting at ⑫ is corollary to active power in the top branch, using input $\hat{Q}_h(t_n)$ and demodulating signal $\tilde{v}_m^{-90}(t_n)$. However, with $\tilde{v}_G(t_n)$ identified and $\tilde{v}_G^{-90}(t_n)$ already available from the quadrature branch, no comparison of candidates is necessary for reactive power because of the quadrature relation established in (7).

Figure 8 shows simulation data of $P(t_n)$ overlaid by the $P_G(t_n)$ (left), as well as grid-frequency sampling of $P(t_k)$ ideally timed to capture the signal envelope compared with down-sampled $P_G(t_k)$ (right).

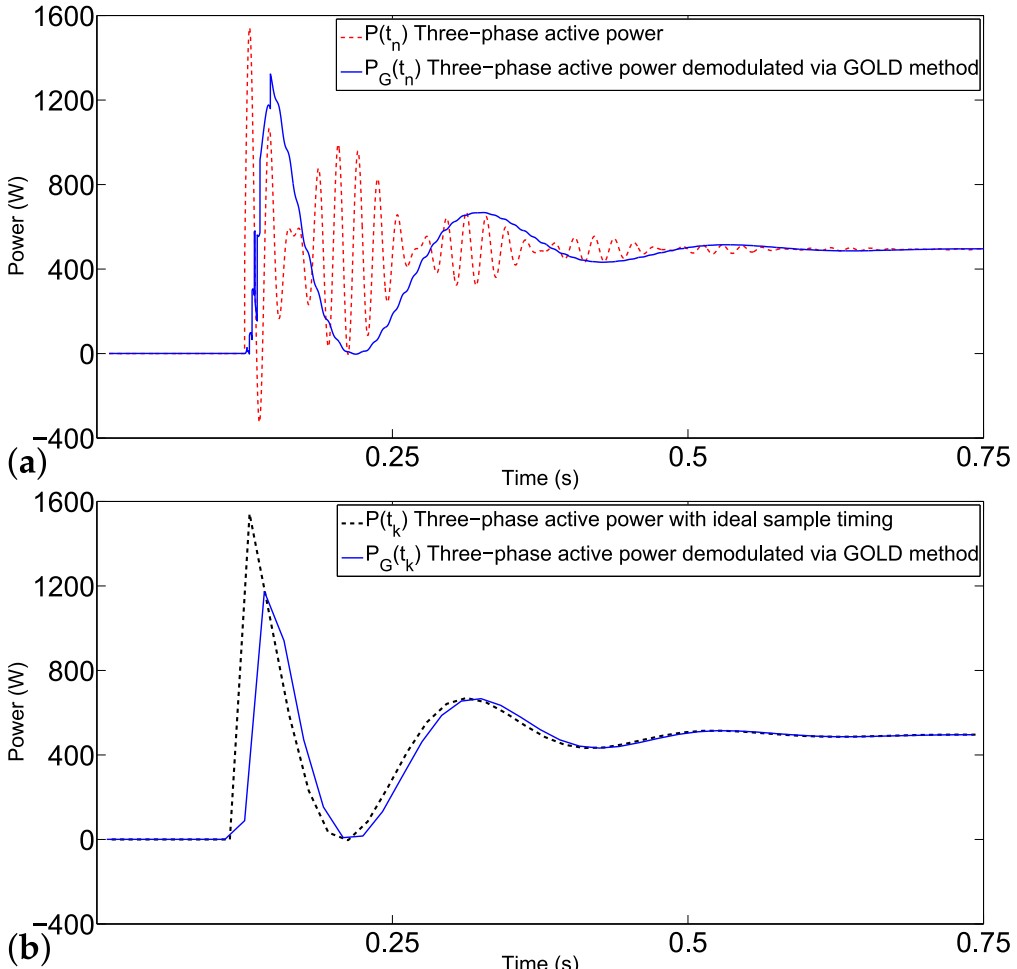

**Figure 8.** Simulated performance of the GOLD method, (**a**) comparing actual computed power disturbance with demodulated power disturbance, and (**b**) comparing of the envelope to down-sampled demodulated power.

## 5. Robustness

### 5.1. Unbalanced Loading

Unbalanced loads present a robustness concern. The imbalances in the reference circuit implementation were assumed to be small enough to be neglected up to this point, but DSB-SC oscillations occur when unbalanced loads are switched in as well. Since demodulation is done on a full three-phase power basis, unbalanced circuits can also be analyzed. As seen in Figure 9 the demodulation succeeds in simulation. For the figure, the reference circuit was made unbalanced by opening the reactive branch on two phases. Note the double-frequency component characteristic of unbalanced loads has not prevented demodulation.

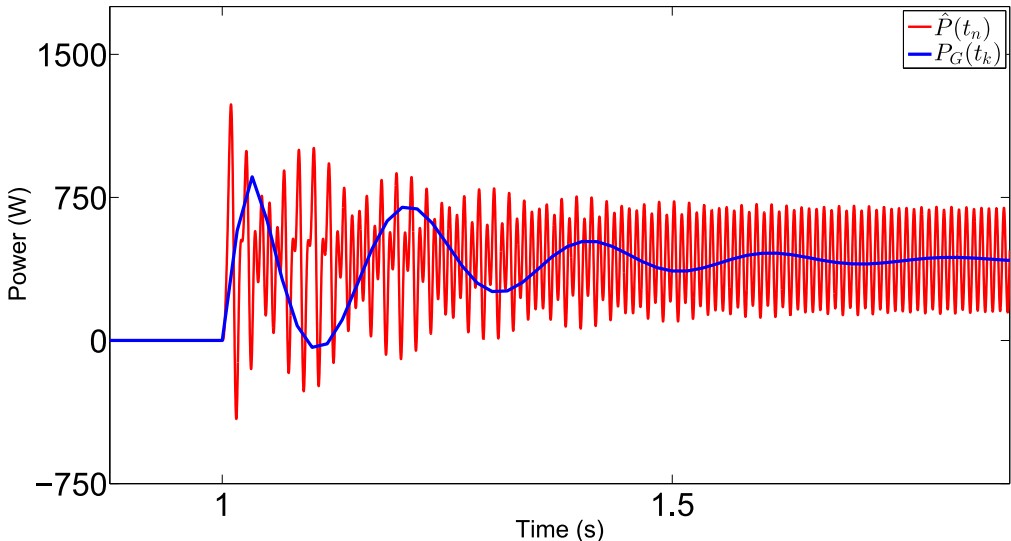

**Figure 9.** GOLD method performance for an unbalanced load: Correct power demodulation $P_G(t_k)$ despite harmonics in instantaneous power $P(t_k)$.

### 5.2. Harmonic Distortion

The more severe aspect of robustness is harmonic distortion. Harmonic distortion most adversely affects the GOLD method algorithm at the band-pass filter.

The choice of $\hat{P}_h(t_n)$ as algorithm input rather than $\hat{P}(t_n)$ limits filter input during steady state: if $P_h(t_n) = 0$ then by combining (10) and (14) the filter input becomes $f_m(t_n) = B^2(t_n)\tilde{v}_m(t_n)\tilde{v}_m^{-90}(t_n)$. At steady state $B(t_n)$ is made up of only what cannot be filtered from the distorted power signal. However, immediately following a transient $B(t_n)$ is dominated by $P_p(t_n) - \hat{P}_p(t_{n-1})$, making band pass filter output $g_m(t_n)$ most appreciable after a transient.

Despite algorithm input choice, signal processing capability is limited by FPGA resources, making admittance of some harmonic distortion inevitable.

To analyze the effect of harmonic distortion, we introduce notation

$$P^{(2)}(f_c, 3f_c) = \sum_{p=a}^{c} i_p(2f_c)v_p(f_c), \tag{18}$$

based on (3), indicating the $H = 2$ harmonic of phase current $i_a, i_b$ and $i_c$ interactions with the $H = 1$ (non-Harmonic) phase voltages $v_a, v_b$ and $v_c$. By application of (8) it is clear that this product contains harmonics with frequencies $f_c$ and $3f_c$ as indicated by $P^{(2)}(f_c, 3f_c)$.

As an update to (10), imperfect filtering leads to these harmonics appearing in

$$\hat{P}_h^{(2)}(f_c, 3f_c) = \hat{P}^{(2)}(f_c, 3f_c) - \hat{P}_p^{(2)}(f_c, 3f_c), \tag{19}$$

the GOLD algorithm input at ④ in Fig 7. Further application of (8) for the product at ⑤ in the same figure begets

$$\hat{P}_h^{(2)}(f_c, 3f_c)\tilde{v}_m(f_c) = P_{Im}^{(2)}(2f_c, 4f_c) + .... \tag{20}$$

Here and forthwith, the ellipsis encapsulates terms not relevant to the discussion such as higher frequency or DC terms.

The same process can be repeated for the $H = 3$ harmonic in the quadrature branch at ⑥:

$$P^{(3)}(2f_c, 4f_c) = \sum_{p=a}^{c} i_p(3f_c)v_p(f_c) \tag{21}$$

$$\hat{P}_h^{(3)}(2f_c, 4f_c)\tilde{v}_m^{-90}(f_c) = P_{Qm}^{(3)}(f_c, 3f_c) + \dots \tag{22}$$

Then at ⑦ the interrelation of the harmonics from (20) and (22) manifests as a grid-frequency component in the band pass filter input:

$$P_{Im}^{(2)}(2f_c, 4f_c)P_{Qm}^{(3)}(f_c, 3f_c) = f_m(f_c) + \dots \tag{23}$$

With this observation, the design problem becomes a trade-off between signal delay from filtering vs. response of band-pass filter output into the desired signal upon switch closure, both constrained by FPGA resources.

The grid-frequency $f_c$ component of $P^{(2)}(f_c, 3f_c)$ from (18) is particularly vexing, because it cannot be removed through filtering since it is the carrier frequency that we aim to identify in our demodulation. However, $v_a(t_n)$, $v_b(t_n)$ and $v_c(t_n)$ present as comparatively undistorted sinusoids at $f_c$, and can again be leveraged given the observation that harmonics in $i_a(t_n)$, $i_b(t_n)$ and $i_c(t_n)$ arise from deterministic processes.

Subtraction of appropriate weightings of $v_a(t_n)$, $v_b(t_n)$, and $v_c(t_n)$ from $P^{(2)}(f_c, 3f_c)$ allows suppression of the $f_c$ component without filtering. The sum of two sinusoids of equal frequency is another sinusoid at the same frequency [46], and the $2\pi/3$ radian phase difference between voltages facilitates construction of a sinusoid fit for canceling the undesired part of $P^{(2)}(f_c, 3f_c)$. This is easily implemented and requires few additional FPGA resources by merging the process with the power calculation:

$$\sum_{p=a}^{c} v_p(t_n)(i_p(t_n) + b_p) = P(t_n) + \sum_{p=a}^{c} b_p v_p(t_n). \tag{24}$$

Determining weightings $b_p$ is achieved by iteration upon steady-state data, seeking values resulting in the minimum-valued DFT bin corresponding to $f_c$.

Removal of the $f_c$ component of $P(t_n)$ allows a higher cut-off frequency for a low pass filter applied to $P(t_n)$ to make $\hat{P}(t_n)$. This filtering further improves robustness against $THD_i$. Filter design is a compromise between minimizing distortion on $\hat{P}(t_n)$ while limiting filter impact on oscillations due to load and circuit dynamics. Filtering of $P(t_n)$ rather than raw current and voltage signals uses fewer FPGA resources, results in less delay of $\hat{P}(t_n)$, and facilitates near immediate band pass filter response to transient by minimizing filter input $f_m(t_n)$ during steady state operation. Critically, a well-designed filter enables the GOLD method to accurately decode power transients even with extreme distortion on the algorithm input.

## 6. Testing and Experimental Results

### 6.1. Hardware Implementation

National Instruments graciously provided a cRIO 9065 microcontroller utilizing a Xilinx Zynq 7000 System on Chip (SOC). The SOC has integrated dual ARM Cortex A9 microprocessor and Z-7020 FPGA. The FPGA facilitates high speed signal processing and full implementation of the GOLD method algorithm as depicted in Figure 10. A three-phase implementation of the reference circuit depicted in Figure 3 was constructed at the SyGMA lab.

A 3-level PWM Grid-Tied Inverter with 6-switch topology was procured from industry partner One-Cycle Control. The inverter switching frequency is 20kHz. The inverter achieves four-quadrant control of power by adjusting magnitude of active and reactive power output via two analog inputs, and toggling supply/consumption of active power or reactive power lead/lag via respective Boolean inputs. It is worth noting that specific electrical parameters for the inverter are irrelevant as the inverter is modelled as a black box with dynamic properties identified from the step response experiments as described in Reference [33]. At rated power of 36kVA, the inverter has Total Harmonic Distortion

of current ($THD_i$) less than 5%, but when operating at low power output $THD_i$ is considerable, as annotated in Table 1 and seen in Figure 1.

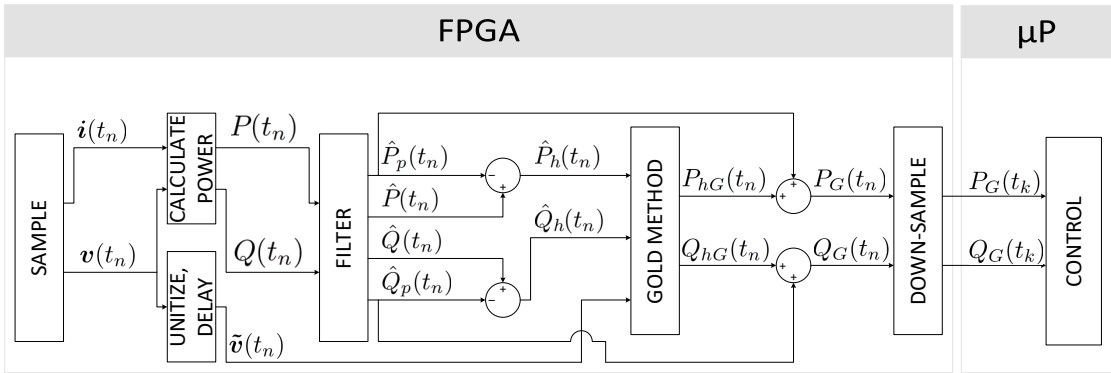

**Figure 10.** Hardware implementation and signal conditioning for the GOLD method algorithm.

$THD_i$ is calculated using

$$THD = \frac{\sqrt{\sum_{H=2}^{\infty} \lambda_H^2}}{\lambda_1},\tag{25}$$

where $\lambda_1$ is the magnitude of the fundamental $f_c$, and $\lambda_H$ indicates magnitudes of harmonics $H * f_c$, $H \in 2, 3, 4....$ To assure that the GOLD algorithm functions throughout the inverter operating range, testing was conducted with the inverter operating at the power output levels listed in Table 1 where signal to noise ratio and $THD_i$ are worst. Power output correlates to the magnitude of the denominator of (25), so when power output is minimized, any harmonics drive $THD_i$ high, nearly to arbitrary extremes. At these power levels total harmonic distortion of voltage ranges from 2.3–2.6%. This use case is presented explicitly to show that the algorithm works even in severe cases, and can thus be expected to function under more conventional circumstances as well.

**Table 1.** Inverter Power Commands and $THD_i$.

| Power (Percent of Nominal) | Power (Watts) | $THD_i$ (%) |
|---|---|---|
| 0.00% | 0 W | 99.4% |
| 0.31% | 113 W | 100.5% |
| 0.63% | 225 W | 143.9% |
| 0.94% | 338 W | 160.6% |
| 1.25% | 450 W | 146.5% |
| 1.56% | 563 W | 118.7% |
| 1.88% | 625 W | 100.9% |
| 2.19% | 788 W | 84.5% |
| 2.50% | 900 W | 71.5% |
| 2.85% | 1013 W | 63.6% |
| 3.13% | 1125 W | 57.2% |

Under these conditions, a distorted oscillation is produced when the reference circuit is switched in while the inverter supplies power. That oscillation is suitable for testing robustness of the GOLD method implemented on the FPGA.

*6.2. Demonstration on Reference Circuit*

Referring back to Figure 4, when switching in the RLC load, the power oscillation should have frequency of approximately 5 Hz. Demonstration of power demodulation in presence of harmonic

distortion is now shown in Figure 11, where indeed the anticipated oscillation of 5 Hz can be seen. The figure shows measurements of distorted active power $P(t_n)$ (top) and reactive power $Q(t_n)$ (bottom) after switch closure, along with each low-pass filtered ($\hat{P}(t_n)$ and $\hat{Q}(t_n)$, respectively) and GOLD method demodulation of the respective filtered signals $P_G(t_k)$ and $Q_G(t_k)$. Figure 11 illustrates the algorithm in action with inverter power output set at 0.94% rated power, the test case at which $THD_i$ is the most severe. Additional testing was done at the inverter power output levels listed in Table 1 with similar results.

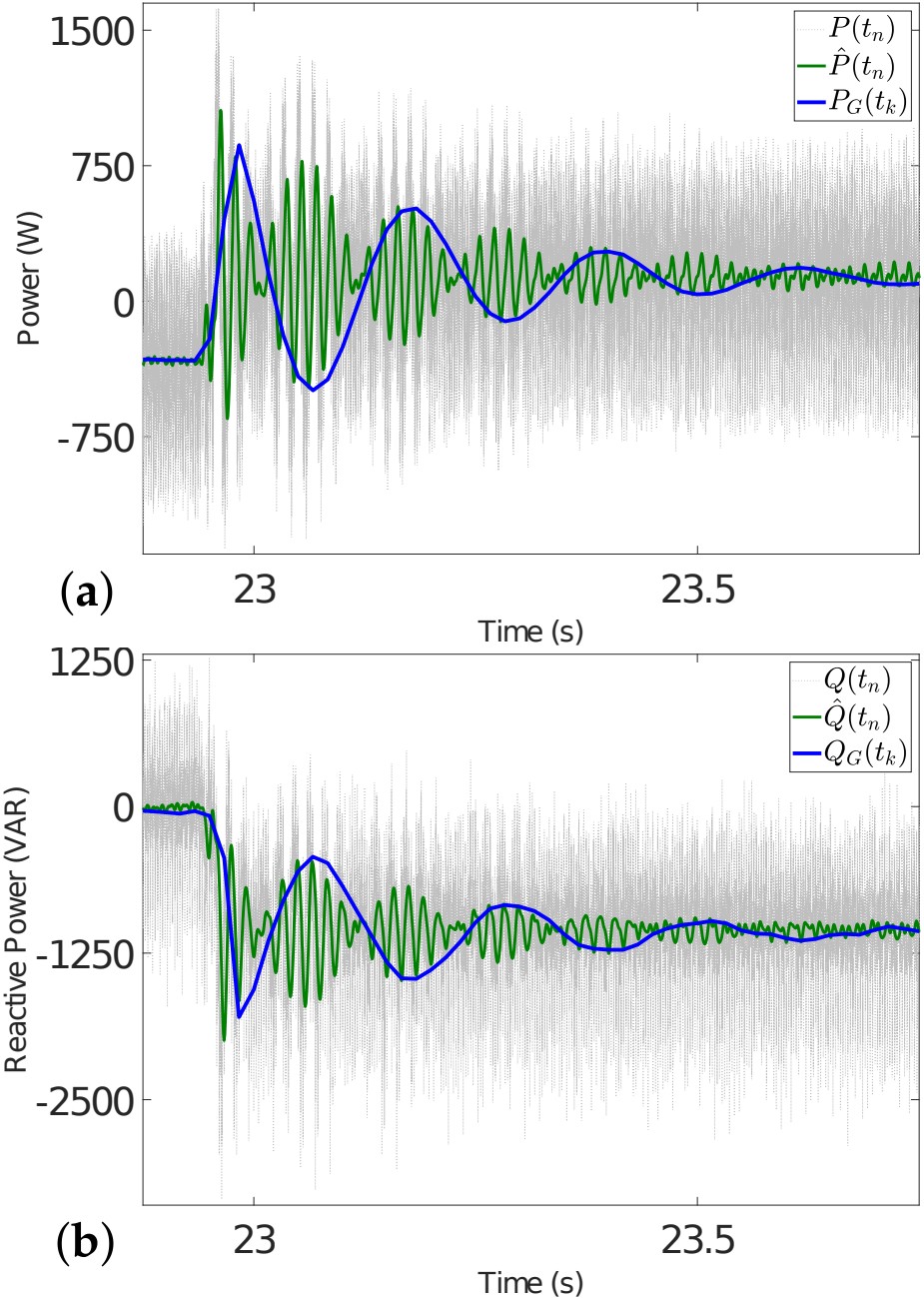

**Figure 11.** GOLD method actual performance, illustrating both (**a**) real and (**b**) reactive power disturbance demodulation.

The GOLD method demodulates the 5 Hz oscillation with fidelity regardless of harmonic distortion on the current supplied by the inverter. Figure 12 shows measurements of distorted active power $P(t_n)$ after switch closure with the reactive branch open on two phases, along with

demodulated power $P_G(t_k)$ and again showing filtered power $\hat{P}(t_n)$ for reference, demonstrating robustness against unbalanced loading despite harmonic distortion.

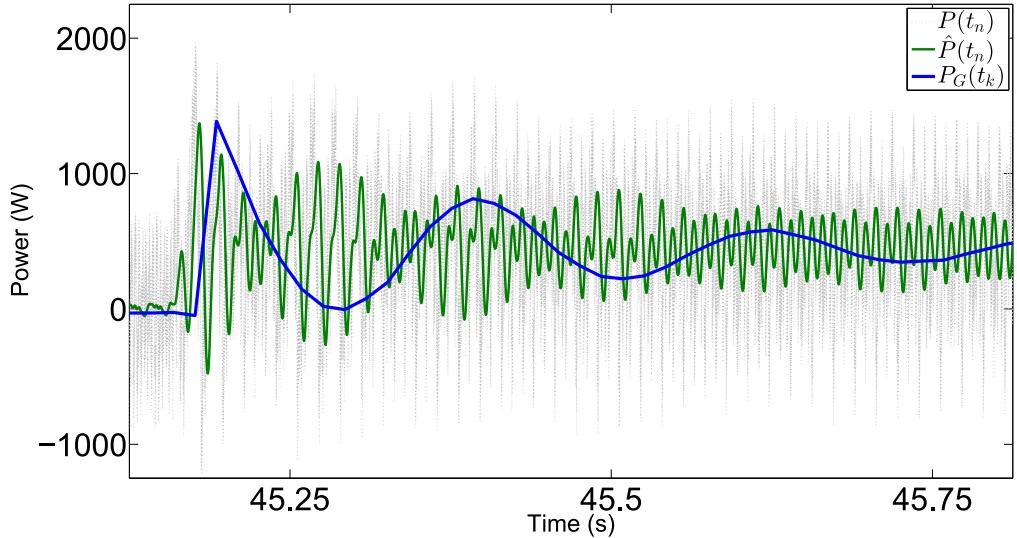

**Figure 12.** GOLD method actual performance for an unbalanced load, illustrating real power disturbance demodulation.

The delay predicted in the simulation shown in Figure 8 is made marginally worse by additional delay from filtering necessary to limit harmonic distortion on the algorithm input. However, the harmonic distortion shown on Figure 11 obscures the transient dynamics severely, making the trade-off of delay for a clean signal acceptable.

*6.3. Improvement in Signal Quality*

The GOLD algorithm operates on power signals, making THD an impractical metric for discussion of the effect of distortion on algorithm signals—At steady-state the power draw of a balanced load has no fundamental frequency, making (25) undefined. However, improvement in signal quality can be quantified with root mean square error of instantaneous active power $P(t_k)$

$$RMSE_{raw} = \sqrt{\frac{1}{K}\sum_{k=1}^{K}[P(t_k) - \overline{P}(t_k)]^2}, \tag{26}$$

compared to the similarly-defined $RMSE_G$ of demodulated power $P_G(t_k)$. In each case the mean is used as estimator at steady state. Recall, per (3), $P(t_k)$ is calculated from raw voltage and current signals. As seen in Table 1, at 0.94% inverter power output, the inverter-supplied current is subject to 160.6% $THD_i$, causing the correspondingly high $RMSE_{raw}$ of 363W; at the same power, $RMSE_G = $ 4.30W, or a reduction in RMSE of 98.8%.

**7. Conclusions and Future Work**

The experimental results presented in this paper show that the GOLD method provides rapid decoding of active and reactive power transients on balanced and unbalanced loads. As can be seen in Figures 11 and 12, the signal decoded by the GOLD method approximates the disturbance envelope within 1.5 cycles of the onset of the disturbance. Simple filtering and signal conditioning allow the algorithm to function despite extreme harmonic distortion on inverter current, that is, $THD_i$ as high as 160.6%, leading to a 98.8% improvement in signal quality. Control of reactive power decoded by the GOLD method is presented in Reference [33] with experiments on active power control ongoing.

**Author Contributions:** Conceptualization, B.T.G.; methodology, B.T.G. and R.d.C.; software, B.T.G.; validation, B.T.G.; formal analysis, B.T.G. and R.d.C.; investigation, B.T.G.; resources, R.d.C.; data curation, B.T.G.; writing—original draft preparation, B.T.G.; writing—review and editing, R.d.C.; visualization, B.T.G. and R.d.C.; supervision, R.d.C.; project administration, R.d.C. All authors have read and agreed to the published version of the manuscript.

**Funding:** This research received no external funding.

**Conflicts of Interest:** The authors declare no conflict of interest.

## Nomenclature

This table of nomenclature is not comprehensive. It is provided to define symbols not explicitly defined elsewhere in the paper, and as reference for symbols that are used outside the context of their initial definition within the text.

| | |
|---|---|
| $f_c$ | Grid frequency |
| $N$ | Number of samples per $f_c$ cycle |
| $\omega_c$ | Angular frequency of voltage input, $2\pi f_c$ |
| $\omega_0$ | Reference circuit natural frequency |
| $t_s$ | Time of reference circuit switch-closure |
| $t_n$ | Discreet time variable with time step $\frac{1}{f_c * N}$ |
| $t_k$ | Discreet time variable with time step $\frac{1}{f_c}$ |
| $v_p$ | Voltage on phase $'p'$ where $p = a, b,$ or $c$ |
| $v_p^{-90}$ | Superscript $'-90'$ indicates $-\pi/2$ radian phase shift on the signal |
| $v_{pp+1}$ | Line voltage, $p = a, b,$ or $c$ and $p + 1 = b, c,$ or $a$, respectively |
| $V$ | Input voltage amplitude |
| $\tilde{v}_m$ | 'Candidate' signal 'm': normalized voltage, may be delayed |
| $\phi_s$ | Voltage phase at time of switch closure. |
| $i_p$ | Current on phase $'p'$ |
| $I_p$ | Steady-state current amplitude |
| $I_h$ | Transient current amplitude |
| $\eta$ | Reference circuit damping attenuation $\frac{R}{2L}$ |
| $\alpha$ | Phase difference between steady-state current and voltage |
| $\beta$ | Phase of three-phase power fluctuation |
| $P$ | Three-phase instantaneous active power |
| $P_p$ | Steady-state component of active power |
| $P_h$ | Transient component of active power |
| $\hat{P}$ | Filtered three-phase instantaneous active power |
| $\hat{P}_p$ | Reconstruction of steady-state component of active power |
| $\hat{P}_h$ | Estimated transient component of active power |
| $P_{hGm}$ | Transient component of active power demodulated by $\tilde{v}_m$ |
| $P_G$ | Decoded active power signal |
| $Q$ | Three-phase instantaneous reactive power |
| $\hat{Q}_h$ | Estimated transient component of reactive power |
| $f_m$ | Band-pass filter input 'm' |
| $g_m$ | Band-pass filter output 'm' |
| $F$ | Filter transfer function. Subscript denotes type: 'LP'-low-pass; 'MA'-moving average; 'BP'-band-pass |
| $B$ | Function representing error due to filtering |
| $H$ | Harmonic order. $H \in 2, 3, 4...$ |

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
