# Peer review of "Demodulation of Three-Phase AC Power Transients in the Presence of Harmonic Distortion"

_energies, doi:10.3390/en13092341_

Round 1

Reviewer 1 Report

This paper proposed an algorithm based on Costas Loop to quickly quantify the phase difference between each candidate signals formed from phase voltages and carrier. Algorithm functionality is demonstrated in the presence of inverter-induced distortion. As a reviewer, this article can be improved by considering the following comments.

The symbols in Figs. 5 and 7 broke the signals. An equal sign is missed at line 105. The arrangement of Figs. 7 and 8 should be reversed. What does it mean of “* denotes 6x zoom on the signal wave-form time axis” at line 188? Which devices of FPGA and microcontroller are used in Fig. 7? Except the authors’ paper [25], there are very few recently published references.

Author Response

Please find responses to all comments in the file attached.

Reviewer 2 Report

The proposed paper titled “Demodulation of Three-Phase AC Power Transients in the Presence of Harmonic Distortion” deals with the signal processing issues concerning the transient effects, real and reactive power flows oscillations, due to load switching and addresses a measuring algorithm for such effects under harmonic distortions. The topic concerned is interesting; however, it is indeed not suitable for publication in its present form.  As the authors state in the conclusion section, … “control of reactive power decoded by the GOLD method is presented in [25] with experiments on active power control ongoing.” To this reviewer, the contribution of the present paper is not significant since the GOLD method has been presented in [25] and this paper does not provide sufficient and integrated technical content demonstrating possible new contributions. Some other reasons and findings are given below for the authors and editors to check.

The paper was not well prepared. The arrangement of technical contents is not acceptable., e.g., in page1/15, about Figure 1, It says …The measurements are taken at the Synchrophasor Grid Monitoring and Automation (SyGMA) Lab at the University of California, San Diego (UCSD) and a comparison is made between grid supplied power (top) and inverter supplied power (bottom).

There is no information about power, only inverter’s currents.

It follows that… on the top of page 2/15, …It is clear that the inverter creates several harmonics on the current and it will be harder to qualify the active and reactive power flow produced by the inverter….

There is no information about inverters, switching methods and the output filtering parameters, etc.  Many similar parts can be found in other sections.

Al the figures presented are too small to see the details. The output current harmonic level or standards of major DG inverters and existing methods for solving the real/reactive power feedback problems published in the literature should be reported. To effectively demonstrate the significance of the proposed algorithm, an experimental test system with a set of real power measuring examples using a 3-phase inverter, a given set of system conditions, and the proposed algorithms is indeed a must. Test scenarios and results must be clearly addressed. The authors claimed that the results presented in the paper show that the GOLD method provides rapid decoding of active and reactive power transients on balanced and unbalanced loads; however, how rapid?

Quantitative analysis is missing; system conditions, key waveforms; e.g., voltages and currents are not presented.

Author Response

Please find responses to all comments in the attached file.

Reviewer 3 Report

This paper presents Demodulation of Three-Phase AC Power Transients in the Presence of Harmonic Distortion. Comments to authors are as follows:
  1. Abstract – authors must expand the acronyms used in the abstract.
  2. The paper is poorly structured and very difficult to follow.
  3. The paper seems to be incomplete, it is recommended to include the experimental work into this paper.
  4. The written language of the paper must be improved.
  5. Conclusions section is too small, authors must elaborate on the conclusions.

Author Response

(The authors gave the same response as above.)

Round 2

Reviewer 2 Report

Dear authors,

Thanks for providing responses to the comments of reviewer 2.

Indeed, there are still a number of comments that required further clarification.

Please see the uploaded file.

Author Response

Thank you for your patience and clarification of previous comments. Please find responses in the attached document.

Reviewer 3 Report

Authors have satisfactorily addressed all the comments raised by the reviewer.

Author Response

The authors extend heartfelt thanks to the reviewer for taking the time to read the manuscript and providing feedback helping to improve it.

Round 3

Reviewer 2 Report

The authors have revised the manuscript based on part of my previous comments; however, there are still some important points required further clarifying and verifying before publication.

Previous Comment_2-1:

Page 1/18, …The measurements are taken at the Synchrophasor Grid Monitoring and Automation (SyGMA) Lab at the University of California, San Diego (UCSD) and a comparison is made between grid supplied power (top) and inverter supplied power (bottom)…

As I commented, in Figure 1, only the voltage and current waveforms are shown. It is not appropriate to state “…a comparison is made between grid supplied power (top) and inverter supplied power (bottom)”.

I suggest the authors to revise it with a clear statement. An example is given below. The authors may use their own way of addressing the comparison target to avoid misleading.

The measurements are taken at the Synchrophasor Grid Monitoring and Automation (SyGMA) Lab at the University of California, San Diego (UCSD) and a comparison is made between the current waveforms of grid supplied power (top) and inverter supplied power (bottom)…

Previous Comment_2-1:

Figure 1. Voltage (blue) and current (red) wave forms when inverter (a) is off and (b) is supplying minimal power, illustrating worst-case harmonic distortion on inverter current.

In Figure 1, the statement, “(b) is supplying minimal power, illustrating worst-case harmonic distortion on inverter current.” is not correct. It should be noted that the harmonic distortion on inverter currents mainly depends on the type of output filter and switching control technology used. It is not strongly related to the control function of inverter, supplying active or reactive power.

What is the actual power command of the Inverter producing the output currents of Figure 1?

It is indeed important to clarify the statement and avoid possible misleading.

Previous Comment_2-3:

Lin, 50-51, There is a significant installed base of inverters that do not use any of these techniques to reduce harmonics [27].

Since the replaced new reference [27] is not accessible. The reviewer can not justify the correctness of this part. Please provide [27] or part of its contents showing the evidence to support the conclusion.

[27] More, A. Inverter Market Share,Size 2019 Exclusivity - by Recent Growth Status, Revenue, Augmentation, key Development Ideas by Market Reports World with Top Players. Market Reports World 2019.

Line, 49-50,.…However, in the United States power quality is regulated by standards tolerant of harmonic distortion [26], limiting adoption of those methods…

The statement is confusing. In [26], the standards concerning the tolerance of harmonic distortion are specified and the techniques reported in [21-25] are normally applied to meet the regulation given in [26].

It is important to clarify the statement: “in the United States power quality is regulated by standards tolerant of harmonic distortion [26], limiting adoption of those methods”.

Previous Comment_2-4:

Line 299-302,  A 3-level PWM Grid-Tied Inverter with 6-switch topology was donated by industry partner One-Cycle Control. The inverter operates at 400Hz and has an LC output filter. At rated power, the inverter has Total Harmonic Distortion of current (THDi) less than 5%, but when operating at low power output THDi is considerable, as seen in Fig. 1.

The basic information the test is not clearly addressed.

The inverter operates at 400Hz (confirmed ?)

What is the rated power of the 3-level PWM Grid-Tied Inverter used?

What is the LC output filter (L & C values) used in the 3-level PWM Grid-Tied Inverter?

What is the control function (i.e., commands, real power  (? W) or reactive power (? Var) of the 3-level PWM Grid-Tied Inverter used (as seen in Fig. 1)?  

Author Response

(The authors gave the same response as above.)

Round 4

Reviewer 2 Report

Dear Authors,

This is my final comment on your manuscript. Two comments are given for your consideration.

Previous Q1:

What is the control command (i.e., real power (? W) or reactive power (? Var) or others) of the 3-level PWM Grid-Tied Inverter used to produce the current waveform shown in Fig.1 (b)? ”

The authors did not answer the above question, (i.e., real power (? W) or reactive power (? Var) or others) of the 3-level PWM Grid-Tied Inverter.

It seems that the authors do not have the related information regarding the experimental setup using in their test cases. For a journal paper, this is indeed not acceptable.

Why this question is important and raised by the reviewer?

If the quantitative information of the power command for the inverter is not stated, it is not possible to realize the power level of so-called “low power output” as stated in the section of experimental tests and results, Line 302-304,

“At rated power of 36kW, the inverter has Total Harmonic Distortion of current (THDi) less than 5%, but when operating at low power output THDi is considerable, as seen in Fig. 1.”

To this reviewer, it is reasonable and a must to give the quantitative information when an experimental test case is presented; rather than describing the setting methods of real and reactive power commands or simply treating it as a black box.

In this study, treating the design of controllers and the details of the inverter circuitry as a black box is acceptable; however, the quantitative information should be addressed.

For the authors reference, the power command for the inverter is estimated at about 550~600VARs.  About 1.5~1.66% of the rated kVA (should be 36kVA). Anyway, the actual quantity and direction of the reactive power should be checked by the authors.

As an example, the authors may try to read the following contents.

Line 307-312,

To assure that the GOLD algorithm functions throughout the inverter operating range,

testing was conducted with the inverter operating at low power output levels where signal to noise ratio and THDi are worst. Power output correlates to the magnitude of the denominator of (22), so when the power output is minimized, any harmonics drive THDi high, nearly to arbitrary extremes. E.g. at 5% rated power THDi was calculated at 103.1%. At the same power level total harmonic distortion of voltage was calculated to be 1.8%.

What is low power output levels actually mean ??%

Final Comment_1

The choice of the operating point (about 1.5~1.66% of the rated 36kVA) to demonstrate the proposed algorithm should be justified.

Final Comment_2

Lin, 302, the rating of 3-phase inverter should be 36kVA, not 36kW.

Author Response

(The authors gave the same response as above.)

Round 5

Reviewer 2 Report

There is still an obvious error to be corrected.

Line 304-306,

...At rated power of 36kVA, the inverter has Total Harmonic Distortion of current (THDi) less than 5%, but when operating at low power output THDi is considerable, as annotated in Table 1 and seen in Fig. 1.

The waveforms shown in Fig. 1 reveals that only reactive power exists.  the real power should be zero. Since the phase difference between the current and voltage is about 90 degrees. That is why I estimated that the command is about 550-600VARs.

The operating commands listed in Table 1 are real power only, the waveforms shown in Fig 1 are not for any of the cases.  

Fig.1 should be replaced with a correct one.

Author Response

Please find response in the attached document.
